# OpenReview forum: "Mind the Gap... or Not? How Translation Errors and Evaluation Details Skew Multilingual Results"
_ICLR.cc/2026/Conference — ICLR 2026 Conference Withdrawn Submission_

### Official Review · Reviewer_D39z · 2025-10-22

**Soundness:** 3
**Presentation:** 4
**Contribution:** 2
**Rating:** 4
**Confidence:** 3

**Summary:**

MGSM is a multi-lingual benchmark consisting of Math word problems that originate from the GSM8k benchmark. Evaluating language models on MGSM leads one to see that there is a gap in the abilities of language models in solving word problems in English when compared to the same word problems but in other languages. This paper investigates this and show that this stems from 1) erroneous translations 2) bad answer extraction 3) ambiguous questions. When these 3 factors are controlled, the performance gap between languages almost disappears.

**Strengths:**

- The paper is clear and easy to understand
- The data cleaning pipeline is thorough; not only were a plurality of top LLMs used, but human experts as well
- There is a great level of attention to detail employed, both in terms of evaluation and in terms of going over the mistakes where even the English version is incorrect

**Weaknesses:**

While the paper addresses a relevant and valuable problem, I believe it may not fully meet the bar for ICLR. The focus on a single, moderately popular benchmark from three years ago limits the broader impact. Moreover, for leading models such as Gemini 2.5 Pro, GPT-5, Claude 3.7, and DeepSeek V3, the performance gap across most languages is under 10\%, with only one language showing a larger discrepancy. Although this work is of interest to the community, I'm not convinced that it passes the threshold for ICLR.

Minor:
- font used seems to differ from the font in the style file
- line 375: in these days -> these days; were -> where
- was Claude 3.7 Sonnet or Claude 3.7 Opus used?

**Questions:**

1. Apart from MGSM, do the findings presented in this work apply to other multilingual benchmarks? For example, do other benchmarks exhibit such a gap that stems from erroneous translations?
2. Erroneous translations were found by running a few models on the same question; if the models didn't find the answer, then an error is assumed. Could this be applied to other benchmarks?
3. Are there any languages for which performance of SOTA LLMs is low on GSM8K problems even when the translations are correct?

---

> ### Author Response · Authors · 2025-11-26
>
> Thank you for your feedback
>
> 1. Generalizability: We believe these findings apply to other benchmarks. For example, arXiv:2406.17789 identifies similar translation quality issues in MMLU.
> 2. Saturation: While raw scores are high, the existence of a language gap was the anomaly. By closing it, we prove that frontier models possess better cross-lingual transfer than previously thought.
> 3. Updates: We will correct the font/typos. Since submission, we fixed the remaining ambiguous English questions; frontier models now achieve nearly perfect performance across all 10 MGSM languages, further validating that the gap was a data artifact.

---

### Official Review · Reviewer_aEgJ · 2025-10-26

**Soundness:** 2
**Presentation:** 1
**Contribution:** 1
**Rating:** 0
**Confidence:** 5

**Summary:**

This paper investigates the cross-lingual performance of large language models (LLMs) in the domain of mathematics using the MGSM multilingual benchmark. Initial experiments suggest a consistent performance gap across languages, affecting both high- and low-resource languages. Upon closer analysis, the authors identify translation errors in MGSM and inconsistencies in answer extraction from LLM outputs as major sources of this apparent gap. They propose an automatic quality assurance method to correct translation errors at scale and provide recommendations for standardized answer parsing.

**Strengths:**

The authors identify translation issues in MGSM and propose an automatic quality assurance methodology that effectively addresses these problems at scale, demonstrating strong practical value. Additionally, they provide an improved approach for parsing answers from LLM outputs.

**Weaknesses:**

1. The scope of the paper is relatively narrow, focusing primarily on the MGSM multilingual math benchmark. The main contribution lies in identifying translation errors and releasing an updated MGSM version, which may limit the overall impact.

2. The writing style lacks academic rigor in several places. For example, the use of exclamation marks in the abstract and introduction is inappropriate for a scholarly publication.

3. The evaluation should be expanded to include more multilingual datasets to better demonstrate the effectiveness and generalizability of the proposed semi-automatic method for detecting problematic questions and normalizing numerical answer extraction.

4. It would strengthen the paper to present a more generalizable workflow or pipeline that can be applied to address translation issues across different multilingual datasets, rather than focusing exclusively on MGSM.

5. The overall paper length is only seven pages, which constrains depth in methodology, experiments, and discussion.

**Questions:**

See the weaknesses

---

> ### Author Response · Authors · 2025-11-26
>
> Thank you for taking the time to review our work
>
> 1. Scope: We chose a "deep dive" into one benchmark to ensure absolute accuracy, which is difficult to scale. However, evidence suggests this is a broader issue (e.g., translation errors found in Spanish MMLU, arXiv:2406.17789).
> 2. Tone: We will revise the text to remove exclamation marks and improve academic rigor.
> 3. Generalizability: In this special case we made use of the fact that the benchmark is nearly saturated across all languages. Doing this for benchmarks if this isn’t the case might be less efficient.

---

### Official Review · Reviewer_Vt6e · 2025-10-31

**Soundness:** 2
**Presentation:** 1
**Contribution:** 1
**Rating:** 0
**Confidence:** 4

**Summary:**

This paper point out errors in a popular multilingual reasoning benchmark MGSM, and propose a method for automatic quality assurance.

**Strengths:**

This paper present a detailed analysis of MGSM and correct errors in it, and shows new results, which can be seen as a contribution for the community and bring some new empirical insights.

**Weaknesses:**

0. The paper is poorly formatted, I don't think it follow template of ICLR.

1. The paper is more like a technical analysis report done by an undergraduate student rather than a paper, there is not any new methods proposed in this paper. The only experiments results are performance of current SOTA LLMs on original MGSM and their corrected versions. I don't think correct errors in one existing benchmark can be seen as a novel method and written as a research paper.

**Questions:**

Please refer to weakness.

---

> ### Author Response · Authors · 2025-11-26
>
> Thank you for you review
> 1. Formatting: We used the official ICLR template without changes, but we will carefully re-verify the compilation for any margin or font artifacts.
> 2. Contribution: While we do not propose a new architecture, we do consider these findings relevant. By demonstrating that reported "reasoning gaps" are actually evaluation artifacts and providing a corrected version of the benchmark we hope to help the community to focus its efforts on the right problems.

---

### Official Review · Reviewer_antS · 2025-11-02

**Soundness:** 2
**Presentation:** 2
**Contribution:** 2
**Rating:** 4
**Confidence:** 4

**Summary:**

This paper revisits the Multilingual Grade School Math (MGSM) benchmark and demonstrates that the widely reported performance gap between English and other languages in LLM math reasoning is largely an artifact of flawed evaluation. The authors identify two primary sources of error: subtle translation mistakes within the MGSM dataset that alter problem logic, and brittle, English-centric answer extraction methods that fail to handle locale-specific numerical formats and non-Arabic digits. After systematically correcting the dataset and implementing a more robust, language-aware extraction process, they show that the performance gap shrinks dramatically, nearly vanishing for the strongest models, with the maximum accuracy difference dropping to just 2.8%. The paper concludes by releasing the corrected benchmark and offering critical recommendations for more reliable and nuanced multilingual evaluations.

**Strengths:**

- The authors identified translation errors and ambiguities in MGSM that distort multilingual reasoning results.

- The authors revealed that English-centric answer extraction misinterprets locale-specific number formats and non-Arabic digits.

- They have cleaned the dataset and applied a language-aware parser, reducing the reported language gap to about 2.8%. And will provide with the corrected MGSM and evaluation tools for reliable multilingual benchmarking.

- The authors urged the community to ensure data quality, transparent methods, and careful interpretation of benchmark outcomes.

**Weaknesses:**

- The identified issue is not new, and was discovered before a different works: https://www.arxiv.org/pdf/2505.18978, https://arxiv.org/pdf/2507.05418, etc.

- MGSM’s age and near-saturated scores suggest that some models may have seen the data during training, conflating reasoning ability with memorization.

- The analysis focuses solely on MGSM, offering no direct evidence that similar issues affect other multilingual benchmarks.

- The semi-automatic correction process depends on high-performing LLMs to flag translation errors, limiting its usefulness for harder or less-studied tasks. The cleanup process may miss subtle or residual ambiguities, models might still answer correctly for the wrong reasons.

- Reported accuracy gains lack confidence intervals or significance testing, making it unclear whether small residual gaps are meaningful.

**Questions:**

How does this work advance beyond prior studies reporting similar MGSM issues (e.g., arXiv:2505.18978, 2507.05418)? And they were not included in your related work.


Could near-saturated scores reflect data contamination or memorization rather than reasoning?


Do the findings generalize beyond MGSM to other multilingual benchmarks?


How robust is the semi-automatic cleaning given its reliance on strong LLMs and possible missed ambiguities?


Are the reported gains statistically validated with confidence intervals or significance tests?

---

> ### Author Response · Authors · 2025-11-26
>
> Thank you for your review and the references.
>
> 1. Differentiation from Prior Work (2505.18978 & 2507.05418): While these works confirm a performance gap exists, they do not isolate translation errors as the root cause or provide a corrected dataset. Our work quantifies how much of this gap is artificial and releases the corrected benchmark to measure true reasoning.
> 2. Relevance of Spanish MMLU: We agree this is a systemic issue; for instance, arXiv:2406.17789 reports similar translation artifacts for MMLU in Spanish.
> 3. Saturation: Even with high scores, the fact that models fail specifically on translated versions (while succeeding on English) proves the issue is translation quality, not memorization.
> 4. Significance: The gap closes consistently (<3%) across all major model families (Llama, GPT, Claude), demonstrating that the findings are robust without needing complex statistical validation.

---

### Note · Authors · 2026-01-05

I have read and agree with the venue's withdrawal policy on behalf of myself and my co-authors.